# Direct synthesis of ordered mesoporous materials from thermoplastic elastomers

Mark Robertson[1], Alejandro Guillen-Obando[1], Andrew Barbour[1], Paul Smith[1], Anthony Griffin [1] & Zhe Qiang[1] ✉

The ability to manufacture ordered mesoporous materials using low-cost precursors and scalable processes is essential for unlocking their enormous potential to enable advancement in nanotechnology. While templating-based methods play a central role in the development of mesoporous materials, several limitations exist in conventional system design, including cost, volatile solvent consumption, and attainable pore sizes from commercial templating agents. This work pioneers a new manufacturing platform for producing ordered mesoporous materials through direct pyrolysis of crosslinked thermoplastic elastomer-based block copolymers. Specifically, olefinic majority phases are selectively crosslinked through sulfonation reactions and subsequently converted to carbon, while the minority block can be decomposed to form ordered mesopores. We demonstrate that this process can be extended to different polymer precursors for synthesizing mesoporous polymer, carbon, and silica. Furthermore, the obtained carbons possess large mesopores, sulfur-doped carbon framework, with tailorable pore textures upon varying the precursor identities.

Ordered mesoporous carbons (OMCs), containing pores with sizes ranging from 2 to 50 nm, have attracted tremendous attention in both fundamental and applied research[1]. These materials have distinct advantages over traditional microporous carbons, including synthetic flexibility in tuning the pore size and matrix chemistry, uniform pore structures with excellent accessibility, and enhanced mass transport within the pore channels[2]. These favorable properties of OMCs enable their broad applications in diverse fields, such as energy storage[3,4], water remediation[5], catalyst supports[6,7], and biomedicines[8]. Early works in OMC synthesis relied on a hard-templating method, which infiltrates carbon precursors into a prefabricated mesoporous material (as the template), followed by crosslinking, carbonization, and template removal[9]. This multiple-step process can be time consuming and costly. Alternatively, OMCs can be prepared by leveraging the self-assembly behaviors of surfactants and/or polymers[2]. For example, soft-templating method is widely employed for OMC fabrication, utilizing the cooperative assembly between a carbon precursor and an amphiphilic polymer/surfactant template[10]. Upon carbonization, the templating agent can be thermally decomposed to open up pores,

while the matrix is converted to a carbon framework. This soft-templating method provides great opportunities for facile and versatile modulation of pore characteristics, such as morphology and pore size, by controlling the nanostructures of assembled precursor/template blends[11]. However, it often requires large amounts of solvent to achieve sufficient mixing. Many research efforts have been directed towards addressing these challenges, such as solid-state synthesis of soft-templated OMCs using mechanochemical self-assembly between tannin-metal complexes and various templates[12]. While effective, the processing requires high energy consumption and pore characteristics are still limited to a narrow range based upon commercially available surfactant templates. Despite the demonstrations of scaled synthesis[13–15], OMCs are still effectively absent in most practical, large-scale systems and commercial market.

Furthermore, large size mesopores are desired for transport and diffusion related applications, such as drug delivery[16,17]. Conventional OMC synthesis using commercial surfactants as the templates is commonly restricted to small pore sizes (e.g. 3–10 nm) due the low molecular weight (MW) nature of templates. Increasing the window of

[1]School of Polymer Science and Engineering, University of Southern Mississippi, Hattiesburg 39406 MS, USA. ✉e-mail: zhe.qiang@usm.edu

accessible pore sizes can be achieved through using high-molecular weight block copolymers (BCP) as templates or introduction of pore expanders, which may require multistep custom synthesis that can be challenging/high-cost at commercial scales[18–20]. As an alternative to templating methods, direct conversion of self-assembled BCPs to OMCs can be achieved through steps of crosslinking and carbonization. In general, these BCPs are composed of a majority block that acts as a carbon precursor and at least one minority block that decomposes during the carbonization process to generate pores. Many works in this area have utilized BCPs containing polyacrylonitrile (PAN) as the majority phase due to its ability to serve as a carbon precursor. For instance, Kopeć et al. demonstrated the synthesis of OMCs from polyacrylonitrile-*block*-poly(butyl acrylate) with varied pore sizes ranging from 5.5 to 21.8 nm by altering the molecular weight of the BCP precursors[21]. Similarly, Zhou et al. employed polyacrylonitrile-*block*-poly(methyl methacrylate) BCPs to produce mesoporous carbon fibers, with pore sizes of 3–10 nm and surface areas up to 503 m$^2$/g[22]. However, PAN-based mesoporous carbons generally exhibit limited degrees of ordering due to high glass transition (~100 °C) and melting temperatures (above 250 °C) of the semi-crystalline PAN segment. Additionally, preparation of PAN-based BCPs usually employs controlled radical polymerization and involves multiple steps, including block extension, which are still challenging to scale up due to the complexity in synthesis.

Some of the most common and inexpensive BCPs are thermoplastic styrenic elastomers (TPEs), which are composed of glassy polystyrene (PS) segments as the minority phase and different types of polyolefins, such as polyethylene, polybutylene, and polyisoprene, as the matrix. TPEs have been widely used in various applications including automotive, medical, construction, electrical, and packaging markets[23], with an estimated global consumption above 6 million metric tons (MTs) per year[24]. Due to the immiscibility between different segments, these TPEs can self-assemble into different nanostructures, including spheres, cylinders, and gyroids[25], in which the aggregated PS domains can serve as physical crosslinks to enhance the mechanical properties of the material[26]. Additionally, these BCPs usually have domain spacings in the range of 20–60 nm, significantly larger than the typical sizes of surfactant micelles. The advantages of low cost, wide availability, and reversible physical crosslinking nature render TPEs to be excellent material candidates for many nanotechnological applications, such as for preparing wearable electronic devices with stretchability and self-healing properties[27,28], piezoresistive blends for electromechanical sensors[29], and shape-memory materials[30].

In this work, we demonstrate an innovative and robust method to synthesize OMCs using nanostructured TPEs as precursors through simple and scalable manufacturing methods, in which polystyrene-*block*-poly(ethylene-*ran*-butylene)-*block*-polystyrene (SEBS) is employed as a model system. Specifically, the polyolefin majority phase (i.e. poly(ethylene-*ran*-butylene), or PEB) can be crosslinked at elevated temperatures in concentrated sulfuric acid through an established method originally developed for the production of polyethylene-derived carbon fibers[31,32]. Upon high temperature pyrolysis under inert atmosphere, the PEB matrix can be converted to a carbon framework, while thermal decomposition of the minority PS phase develops ordered mesopores. Additionally, the use of sulfuric acid leads to the incorporation of sulfur heteroatoms in the carbon matrix. As a result, our approach can directly synthesize large-pore, heteroatom-doped OMCs from TPE precursors. This work systematically investigates how the chemical compositions and nanostructures of TPEs evolve after each processing step, including crosslinking, calcination, and carbonization. We also demonstrate that our method can be extended to other SEBS systems with varying molecular weights and chemical compositions, confirming its generalizability to obtain ordered mesoporous materials with distinct pore textures. Furthermore, SEBS-derived OMCs can be used as a hard template for synthesizing large-pore ordered mesoporous silica. Our approach is simple and advantageous toward scaled production of ordered mesoporous materials with large pores, which can be complementary with existing OMC synthesis platforms. These materials are particularly promising for applications that would demand large mesopores for facilitating mass transport of guest molecules and a heteroatom-doped carbon matrix, such as drug delivery and micropollutant removal from aqueous media.

## Results

### Sulfonation-enabled crosslinking of SEBS

The schematic illustration of OMCs synthesis from thermoplastic elastomer (TPE) precursors is shown in Fig. 1a, using a sulfonation-based chemistry for crosslinking the poly(ethylene-*ran*-butylene) (PEB) matrix to enable its efficient conversion to a carbon precursor. Specifically, commercially available, bulk polystyrene-*block*-poly(ethylene-*ran*-butylene)-*block*-polystyrene (SEBS) powders are first treated by thermal annealing at 160 °C for 12 h for obtaining long-range ordering in their nanostructures. The polymer is submerged in concentrated sulfuric acid and reacted at 150 °C to selectively crosslink the olefinic block (PEB). After crosslinking, the polymer can be calcinated under an inert atmosphere to selectively decompose the PS minority phase for producing mesoporous polymers or exposed to temperatures above 600 °C for obtaining OMCs. Compared to conventional soft-templating and direct pyrolysis methods for OMC synthesis, several noteworthy advantages in our system include the wide availability and low cost of raw materials, broadly tunable pore characteristics through precursor selection, and easy and scalable processing steps. As a note, multiple SEBS precursors are used throughout this work which will be referred to as SEBSX, where X denotes the molecular weight of the precursor divided by a factor 1000. Detailed descriptions of each precursor, including molecular weight and polydispersity can be found in the experimental section.

In our method, sulfuric acid was employed as the chemical reagent to crosslink SEBS, relying on the diffusion of sulfuric acid within the SEBS powders, and no dissolution of polymers occurred throughout the entire process (Supplementary Fig. 1). The acid can effectively react with both majority and minority phases at 150 °C, and the distinct reactions for PS and PEB blocks are demonstrated in Fig. 1b. Crosslinking polyolefins through sulfonation of the polymer backbone has been established in the literature for several model precursors, including polyethylene (PE) and polypropylene (PP)[33–36]. The crosslinking reaction proceeds through multiple mechanisms that occur in tandem throughout the sulfonation process. Initially, sulfonic acid groups are introduced to the polymer backbone, which is followed by elimination to form double bonds. These double bonds react through further additions and dissociations, consequently forming radical species that crosslink the polymer chains through intermolecular radical-radical coupling. As demonstrated in previous reports, sulfonation-enabled crosslinking method leads to a high carbon yield of PP and PE, up to 70 wt%[32] Additionally, sulfonation of PS can lead to the addition of sulfonic acid groups on the aromatic ring of the repeat unit, where the *para* position is typically preferred (Fig. 1b)[37].

The reaction progress can be monitored through understanding the changes in mass gain and gel fraction of the sulfonated product as a function of time (Fig. 1c). The SEBS118 polymer steadily increases its mass with longer reaction times, reaching a plateau after approximately 4 h of reaction at 150 °C. A similar trend is observed in the gel fraction of the material, which is determined by washing sulfonated SEBS118 in hot toluene (at 85 °C for 12 h) to remove un-crosslinked fractions. It was found that the gel fraction reached a plateau of 88 wt% after 4 h reaction. The changes in chemical composition of the polymer during the sulfonation reaction can be further elucidated through the Fourier transform infrared (FTIR) spectra as shown in Fig. 1d. It has

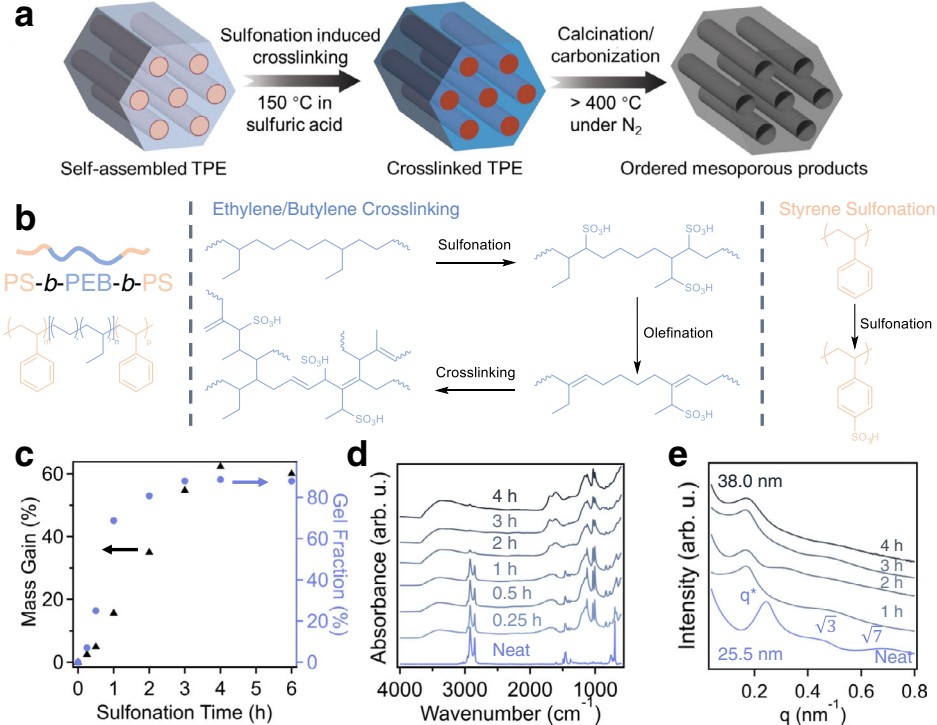

**Fig. 1 | Sulfonation-induced crosslinking reaction of SEBS. a** Schematic illustration of thermoplastic elastomer undergoing a sulfonation induced crosslinking reaction and being converted into a carbon precursor. The precursor is pyrolyzed in inert atmospheres to produce mesoporous products. **b** Crosslinking reaction schemes for both PEB and PS blocks of the SEBS118 precursor. **c** Plot of mass gain and gel fraction of the SEBS118 polymer as a function of sulfonation reaction time. **d** FTIR spectra of SEBS118 as a function of sulfonation time. **e** SAXS patterns of SEBS118 as a function of crosslinking time at 150 °C. Source data are provided as a Source Data file.

previously been reported that the sulfonation of homopolymer PS could occur very rapidly at elevated temperatures in concentrated sulfuric acid[37]. This rapid PS functionalization was also observed in our system, which is evidenced by a prominent vibration associated with the disubstituted aromatic rings of the PS block at 1006 cm⁻¹. At short reaction times (0.25 h, 0.5 h, and 1 h) this vibration is dominant, while the alkyl stretching vibrations associated with the PEB block at 2851 cm⁻¹ and 2920 cm⁻¹ only diminish slightly. This result indicates that the primary reaction occurring at short timescales (within the first hour) is the sulfonation of PS segments. However, the relative intensity corresponding to the alkyl stretching vibrations begins to decrease more significantly after 1 h of sulfonation, and bands associated with the addition of sulfonic acid groups (1033 cm⁻¹) and alkenes (1615 cm⁻¹) within the backbone become more present. These results represent that the progress of the PEB matrix crosslinking has slower kinetics than PS sulfonation. The bands corresponding to the addition of sulfonic acid functional groups to the polymer backbone (1033 cm⁻¹) and the aromatic ring of polystyrene (1006 cm⁻¹) are compared in Supplementary Fig. 2 to provide a qualitative understanding of the reaction progress of the distinct blocks. While the band at 1033 cm⁻¹ is present at short reaction times, it is less prominent than the peak at 1006 cm⁻¹ until 1 h of sulfonation. This result further confirms that the sulfonation of the olefinic backbone has slower reaction kinetics than the PS parts. As reaction time increases over 1 h, intensity of both bands increases, and the presence of alkenes become more prominent, until 4 h of reaction time. Further extending the sulfonation time results in a spectrum that remains nearly constant, indicating the completion of the reaction (Supplementary Fig. 3) of SEBS118 after 4 h at 150 °C in concentrated sulfuric acid. Additionally, titration experiments were performed on sulfonated SEBS118 samples to determine the amount of sulfonic acid groups on the polymer backbone as a function of crosslinking time. This was accomplished following an

established method[38], which introduced the crosslinked SEBS118 in 0.1 M sodium chloride (NaCl) solutions for 48 h to exchange the protons of the sulfonic acid with sodium ions. The solution in which the material was soaked could then be titrated using a 0.026 M NaOH solution to determine the concentration of acid present. Supplementary Fig. 4 depicts the sulfonation degree of SEBS118 samples as a function of crosslinking time, corresponding to the percentage of repeat units in the polymer that contain a sulfonic acid group. This result agrees well with the general reaction conversion trend observed in the mass gain and gel fraction studies. The sulfonation degree increases to 14% after 3 h of reaction time and then decreases slightly to 12%; further olefination and crosslinking of PEB can lead to reduced amount of the sulfonic acid groups on the polymer backbones as shown in Fig. 1b. For comparison, other works which sulfonated SEBS for water desalination membranes demonstrated roughly 4% of the repeat units were sulfonated after 5 h of reaction at 50 °C[38]. The significant increase in the sulfonation degree of SEBS resulting from our process is due to the increased reaction temperature which enables the sulfonation of both PEB and PS blocks, whereas the PEB block cannot be sulfonated at low temperatures (e.g. ~50 °C). From these results, it is important to note that the sulfonation-enabled crosslinking kinetics of SEBS was significantly faster than the semicrystalline counterparts (e.g. homopolymer PP and PE), in which the crystalline regions can limit the progress of the acid diffusion and sulfonation reaction. The SEBS used in this study is amorphous (Supplementary Fig. 5), providing an important mechanism for facilitating the crosslinking reaction and enabling significantly shorter reaction times for bulk sample crosslinking. To illustrate the importance of the elevated reaction conditions used in this process, the sulfonation reaction was also performed at 85 °C and 125 °C for up to 6 h. Mass gain, gel fraction, and FTIR results are presented in Supplementary Fig. 6. Both temperatures exhibit slower kinetics than the sulfonation at 150 °C and

also demonstrate lower plateau values of mass gain and gel fraction. Sulfonation at 85 °C and 125 °C only achieve ~40% mass gain over 6 h compared to 60% for SEBS118 sulfonated at 150 °C. Similarly, the gel fractions of SEBS118 sulfonated at 85 °C and 125 °C were approximately 12% and 60%, respectively. The reduced gel fraction in comparison to the 150 °C reaction condition suggests that lower temperature sulfonation reactions result in reduced degrees of crosslinking. Furthermore, the FTIR spectra in Supplementary Fig. 6b and d indicate a reduced presence of the characteristic bands (sulfonic acids: 1033 cm$^{-1}$ and 1006 cm$^{-1}$, alkenes: 1615 cm$^{-1}$) associated with the crosslinking reaction, in addition to the retention of the alkyl stretching vibrations (2851 cm$^{-1}$ and 2920 cm$^{-1}$) indicating an incomplete reaction. These results suggest that sulfonation temperature of SEBS is an important process parameter to control their crosslinking kinetics.

The effects of the sulfonation reaction on the nanostructure of SEBS118 can be determined using small angle x-ray scattering (SAXS), as shown in Fig. 1e. The neat SEBS118 has a primary ordering peak corresponding to a domain spacing of 25.5 nm, along with higher ordering peaks at ratios of $1 : \sqrt{3} : \sqrt{7}$ with respect to the primary peak position, indicating a hexagonally packed cylindrical morphology. Interestingly, we found that the nanostructure of SEBS is altered almost immediately upon exposure to the sulfonating agent at 150 °C, evidenced by a rapid increase in their domain spacing. Specifically, Supplementary Fig. 7a depicts SAXS patterns of samples at early times in the reaction progression, from 1 to 10 min of reaction. The domain spacing increases rapidly after 3 min of reaction to 38 nm and remains virtually constant throughout 4 h of reaction (Supplementary Fig. 7b). Additionally, the scattering patterns were fit to model scattering functions which included a flexible cylinder form factor to account for scattering contributions from the size and shape of the minority cylindrical PS domains. As shown in Supplementary Fig. 7c, a similar trend to the domain spacing evolution was observed where the cylinder diameter increased rapidly at short time scales from 16.6 to 22.0 nm within 3 min of reaction, and then gradually increased throughout the reaction to 24.0 nm. These results further confirm that the nanostructure is established at very short reaction times and is only altered slightly at extended reaction times. Notably, comparing the increase in cylinder diameter throughout the reaction (~7.4 nm) to the increase in domain spacing (~12 nm) suggests that PS domain expansion as a result of the sulfonation reaction is the primary contributor to the altered nanostructure. As the sulfonation reaction progresses, PEB crosslinking and the presence of ionic groups on polymer backbones can significantly hinder the polymer chain mobility for structural rearrangement, kinetically trapping the morphology of SEBS after relatively short sulfonation. It is noted that the SEBS nanostructures changed towards a slightly reduced degree of ordering with extended reaction times. The full width at half maximum (FWHM) of the primary peak within the scattering patterns was analyzed and provided in Supplementary Fig. 7d. Within the first minute of reaction, the peak broadens significantly as rearrangement of the nanostructure occurs. Subsequently, the FWHM decreases sharply after 3 min of reaction as long-range ordering is re-established. In contrast to the domain spacing and cylinder diameter which remained nearly constant as the reaction progressed, a slight increase is observed in the FWHM with SEBS118 sulfonation progress. As shown in Fig. 1e, the higher ordering peaks became less distinguishable, which further suggests a possible loss in the degree of ordering in the crosslinked polymer. The reduced ordering of BCPs has been commonly observed in many systems, upon simultaneous reactions and nanostructural rearrangement[39,40]. It is important to note the sulfonation reaction would effectively alter the chemical composition of both PS and PEB blocks, thus changing the volume fraction of minority phase. With 14% degree of sulfonation observed in SEBS118 samples (Supplementary Fig. 4), the volume fraction of PS could increase up to 25%, suggesting that cylindrical phase should be maintained throughout the crosslinking

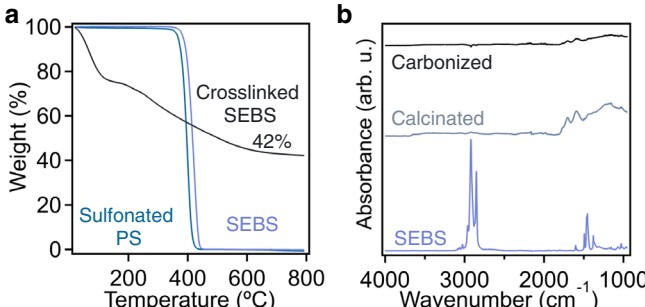

**Fig. 2 | Conversion of SEBS to ordered mesoporous materials through pyrolysis. a** TGA results up to 800 °C in N$_2$ atmosphere for neat SEBS118, sulfonated homopolymer polystyrene, and crosslinked SEBS118. **b** FTIR spectra of neat SEBS118, calcinated SEBS118, and the carbonized product. Source data are provided as a Source Data file.

process of nanostructured SEBS. Particularly, the SEBS118 nanostructure was kinetically trapped within less than 10 min of sulfonation reaction, further limiting the possibility of an order-to-order transition to occur.

## Fabrication of ordered mesoporous materials from sulfonated SEBS

After reacting with concentrated sulfuric acid for 4 h, the polymer matrix is sufficiently crosslinked, allowing its conversion to mesoporous materials with significant char yield. Figure 2a depicts thermogravimetric analysis (TGA) results of neat SEBS118 polymer, sulfonated PS, and SEBS118 after 4 h of crosslinking. Neither the neat polymer, nor sulfonated PS exhibit any carbon yield above ~400 °C in a N$_2$ atmosphere, while SEBS118 crosslinked at 150 °C for 4 h results in roughly 42% residual mass, suggesting a carbon yield of 67 wt% in comparison to the initial mass of the polymer prior to obtaining the mass gain from sulfonation. From these results, it can be found that (1) the sulfonation induced crosslinking is required to produce carbon from the SEBS118 polymer and (2) sulfonation of PS proceeds solely through the reaction with the aromatic ring of the repeat unit, which does not yield carbon products upon pyrolysis. Notably, the sulfonation reaction occurs at the aromatic ring of the PS repeat units, and sulfonic acid groups could undergo crosslinking with other sulfonic acid moieties to produce intermolecular sulfone bridges at elevated temperatures[41]. Although this effectively crosslinks the polymers, these crosslinking sites are still thermally labile and can decompose upon pyrolysis rather than producing carbon. For instance, Maranesi et al. have investigated the mechanism of thermally driven sulfone crosslinking between sulfonic acid moieties in sulfonated poly(ether ether ketone) (PEEK)[42]. Across many thermal treatments of the sulfonated PEEK materials, that could be seen as analogous to the crosslinking reaction used in this work, no considerable residual mass was present after exposure to 800 °C in N$_2$. This suggests that, while the PS domains (~20 vol% in SEBS118) may undergo intermolecular crosslinking through this mechanism, it is still insufficient to produce carbon and will not contribute to the carbon yield of the OMCs. To further confirm these results, SEBS118 was sulfonated at 85 °C for 12 h to sulfonate both the PEB and PS blocks of the polymer without inducing crosslinking. The FTIR spectrum in Supplementary Fig. 8a depicts bands associated with the sulfonation of both PEB and PS blocks (bands at 1033 cm$^{-1}$ and 1006 cm$^{-1}$, respectively). Additionally, a broad band ranging from 1293 to 1111 cm$^{-1}$ is a convolution of vibrations associated with sulfonic acids, as well as sulfones. However, despite the sulfonation of these materials, the TGA thermogram in Supplementary Fig. 8b indicates that no carbon is yielded after carbonization at 800 °C in a N$_2$ atmosphere, suggesting that any crosslinking that may occur within sulfonated PS segments did not produce carbons upon pyrolysis.

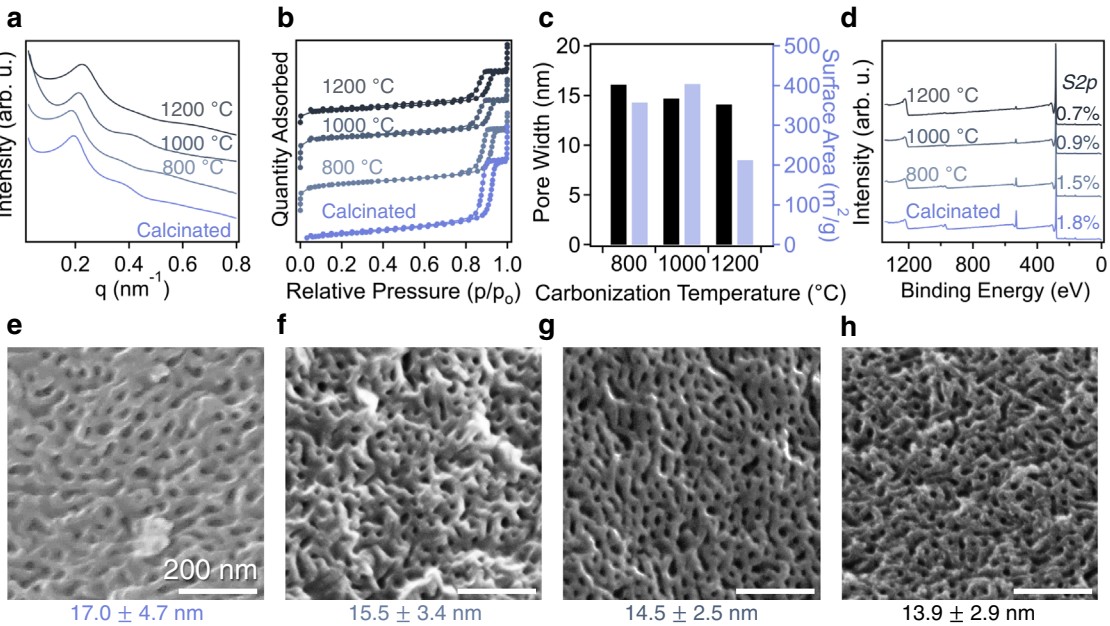

**Fig. 3 | Characterization of SEBS-derived ordered mesoporous materials.**
**a** SAXS patterns for ordered mesoporous polymer and SEBS118-derived OMC carbonized at 800 °C, 1000 °C, 1200 °C. **b** Nitrogen adsorption isotherms which exhibit type IV isotherms, indicating the presence of mesopores. For clarity, the isotherms have been shifted in the positive Y-direction. **c** Pore width calculated through NLDFT models for carbon slit pores at 77 K and the Brunauer-Emmett-Teller (BET) surface area at each carbonization temperature. **d** XPS survey scans of each material indicating the presence of oxygen and sulfur atoms doped within the carbon framework. The inset sulfur content is provided in at%. SEM micrographs of (**e**) mesoporous polymer and mesoporous carbon carbonized at (**f**) 800 °C, (**g**) 1000 °C, (**h**) 1200 °C. Source data are provided as a Source Data file.

By calcinating crosslinked SEBS at 400 °C for 3 h, the minority PS domains were selectively decomposed, leading to the formation of a mesoporous polymer matrix. The effect of the calcination process on the chemical identity of the polymer is exhibited in Fig. 2b. The removal of PS segments can be confirmed by the diminished bands associated with the alkyl stretches of the polymer backbone, as many were reacted to form crosslinks during the sulfonation process, and the alkene stretching vibration at 1603 cm$^{-1}$ is present as a result of the formation of double bonds during the crosslinking reaction. Additionally, the secondary band at 1704 cm$^{-1}$ can be attributed to the presence of various oxygen containing functional groups, such as ketone and aldehydes, that are a result of sulfonation crosslinking reactions. Similarly, the broad band from 1000 to 1500 cm$^{-1}$ is a result of multitudes of sulfur and oxygen containing functional groups installed into the polymer network through crosslinking. The prevalent functional groups indicate that the mesoporous material contains polymer characteristic after the calcination process. After carbonizing at 800 °C, no distinct vibrations are present in the FTIR spectrum, due to the absence of functional groups, which is consistent with other previously reported mesoporous carbon synthesized from soft-templating method[13].

Figure 3 depicts the structural characteristics of SEBS118-derived mesoporous materials (both calcinated and carbonized) which were produced from a precursor that was crosslinked at 150 °C for 4 h, which are summarized in Table S1. After calcination at 400 °C for 3 h, the domain spacing decreases to 32.7 nm (Fig. 4a) and a typical type IV nitrogen adsorption isotherm (Fig. 4b) was observed, confirming the formation of ordered mesoporous structures. The averaged pore size distribution of the mesoporous polymer determined by non-local density functional theory (NLDFT) modeling is approximately 16.1 nm in diameter, and the BET surface area is around 133 m$^2$/g. It is worth noting that the reaction time directly impacts the nanostructure of the porous carbon product. Shorter sulfonation times are still sufficient

for producing relatively well-ordered porous carbon materials as illustrated in Supplementary Fig. 9. Specifically, nitrogen physisorption isotherms (Supplementary Fig. 9a) and pore size distributions (Supplementary Fig. 9b) confirm the presence of ordered mesopore structures. Samples which were crosslinked for 1 h, 2 h, and 3 h demonstrated a gradual increase in the averaged pore size from 14.1 nm (sulfonated for 1 h) to 15.6 nm (sulfonated for 3 h). These results suggest that SEBS118-derived OMC can have process-tunable pore textures, enabling controlled pore sizes by varying crosslinking conditions. The TGA thermogram in Supplementary Fig. 9c reveals that increased reaction times are required for maximizing the carbon yield of the material after carbonization. Samples sulfonated for 1 h exhibited only 12 wt% yield (from sulfonated samples) after exposure to 800 °C under N$_2$, which increased to 26 wt% and 34 wt% after reaction for 2 h and 3 h, respectively. After 4 h, the yield is maximized at 42 wt%, and this condition was considered optimal for further studies. After carbonization at 800 °C, the domain spacing of the sample sulfonated for 4 h slightly increases to 33.9 nm, which shrinks to 29.4 nm and 27.9 nm by increasing pyrolysis temperature to 1000 °C and 1200 °C, respectively. Additionally, the SAXS patterns of these samples all exhibit secondary ordering peaks, indicating the presence of long-range ordering within the hexagonally packed cylindrical morphology. The emergence of these high ordering peaks in the OMCs, compared to crosslinked samples, is likely due to the enhanced scattering contrast between pore voids and the carbon/polymer framework. In comparison, most mesoporous carbon materials derived from PAN-based BCPs typically have a lower degree of ordering, only exhibiting a broad primary peak[21,43]. The changes in domain spacing upon carbonizing at different temperatures correspond well to the pore size distributions calculated from nitrogen sorption isotherms in Fig. 3b, c. The pore size distributions of all SEBS118-derived OMC samples are included in Supplementary Fig. 10. Specifically, the SEBS118 derived OMCs exhibited the averaged pore sizes of 16.1 nm,

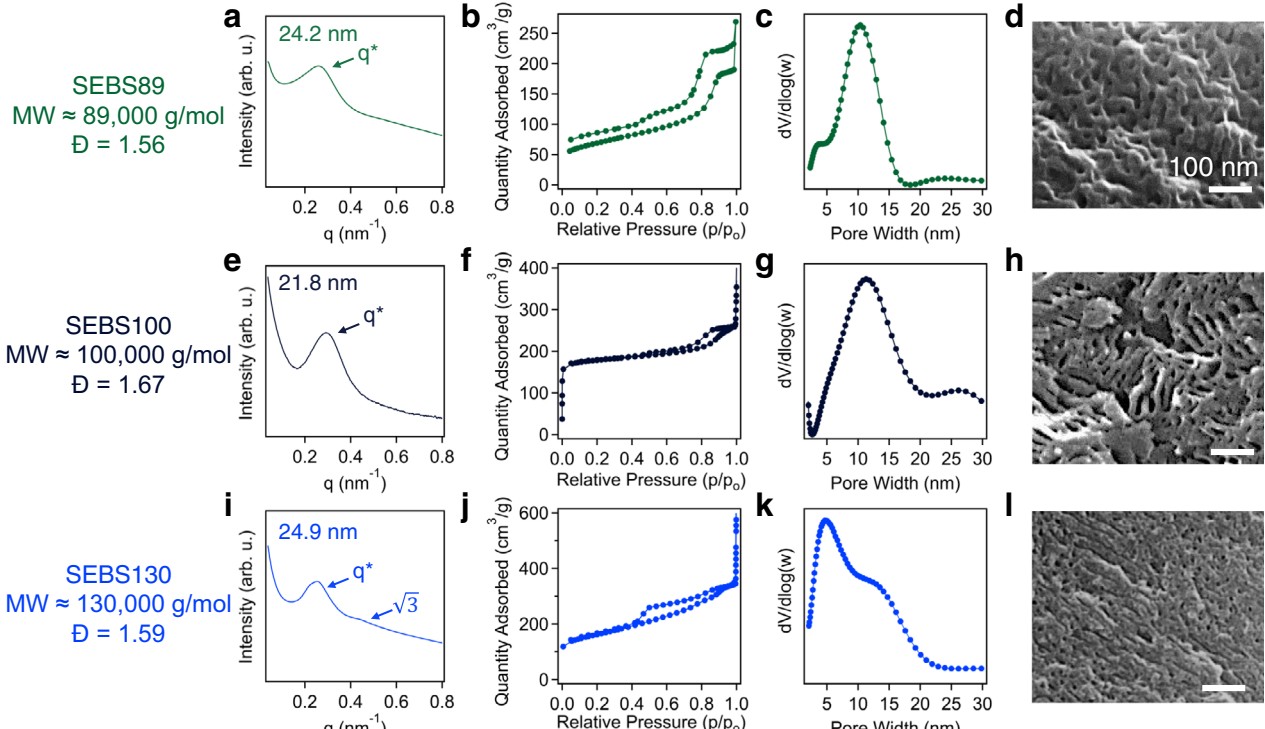

**Fig. 4 | Characterization of OMCs-derived from multiple precursors. a** SAXS profile, (**b**) nitrogen sorption isotherm, (**c**) pore size distribution and (**d**) SEM image of OMC pyrolyzed from SEBS89. **e** SAXS profile, (**f**) nitrogen sorption isotherm, (**g**) pore size distribution and (**h**) SEM image of OMC produced from SEBS100. **i** SAXS profile, (**j**) nitrogen sorption isotherm, (**k**) pore size distribution and (**l**) SEM image of OMC produced from SEBS130. Source data are provided as a Source Data file.

14.7 nm, and 14.1 nm, when they were carbonized at 800 °C, 1000 °C, and 1200 °C, respectively. Notably, when comparing the calcinated mesoporous polymer to the SEBS-OMC carbonized at 1200 °C, only ~9% shrinkage was observed in pore diameters, which is significantly lower than surfactant-templated mesoporous materials (~48%). Furthermore, BET surface areas of these OMC materials are 357 m²/g (carbonized at 800 °C), 404 m²/g (1000 °C), and 212 m²/g (1200 °C) (Fig. 3c). These values are similar to mesoporous materials fabricated using PAN-based BCPs as the precursors[22,44,45], but lower in comparison to Pluronic-templated OMCs[46]. Specifically, one of the most commonly produced soft-templated mesoporous carbon contains pores that are 3.1 nm in width with a surface area of 720 m²/g[10]. While SEBS118-derived mesoporous carbon exhibits a reduced surface area, its much larger mesopore size can be advantageous for mass transport and separation of large-size guest molecules[47]. Additionally, scanning electron microscopy (SEM) images in Fig. 3e–h confirm the uniform pore sizes of these samples, where the darker regions represent the mesopores produced from decomposition of the PS phase, and the brighter regions are the polymer or carbon matrix. The pore size determined from these SEM images is approximately 17.0 nm (calcinated sample), 15.5 nm (carbonized at 800 °C), 14.5 nm (1000 °C), and 13.9 nm (1200 °C), which are consistent with results from SAXS and physisorption measurements. TEM images for SEBS118 carbonized at 800 °C are also provided in Supplementary Fig. 11, further confirming the ordered cylindrical morphologies of the resulting OMC.

In addition to the ability to fabricate large-pore OMCs from SEBS using our process, the sulfonation reaction imparts additional functionality into the final mesoporous product through doping the carbon framework with sulfur heteroatoms. Generally, doping heteroatoms into carbons can enhance their utility in many applications[48,49]. For instance, the presence of sulfur in carbon sorbents has been demonstrated to enhance their CO₂ capture capacity through providing enhanced interactions between the sorbent and the gas molecule[50–52].

Similarly, Lin et al. synthesized ordered nitrogen, sulfur, and oxygen doped mesoporous carbons using cysteine as a carbon precursor[53], demonstrating that the presence of sulfur in carbon framework was critical in enhancing the performance of OMCs for oxygen reduction reactions. Conventional templating methods either require the use of heteroatom-containing carbon precursors or additional materials (e.g. dopants) to incorporate heteroatoms into the carbon framework[54–58]. PAN-based BCP can result in nitrogen-doped mesoporous carbon with nitrogen content up to 13 at%[22], enabling their exceptional performance for many applications, such as fast-charging batteries[59], supercapacitors[60], and water desalination[61]. Sulfonation-enabled crosslinking of PE and PP can result in high sulfur contents, approaching 3 at% as determined through X-ray photoelectron spectroscopy (XPS) in previous reports[62]. For our system, XPS results (Fig. 3d) of the calcinated mesoporous polymer, and the SEBS118-derived OMCs carbonized up to 1200 °C, indicate the presence of sulfur doping in the mesoporous products, the values of which can be found in Table S1. Specifically, the mesoporous polymer and SEBS118-OMC carbonized at 800 °C exhibit sulfur contents of 1.8 at% and 1.5 at%, respectively. Increasing carbonization temperature to 1000 °C and 1200 °C decreases the sulfur content to 0.9 at% and 0.7 at% as the heteroatoms are eliminated from the framework at elevated temperatures. Additionally, *C1s* (carbon), *O1s* (oxygen), and *S2p* (sulfur) high resolution XPS scans for each material can be found in Supplementary Fig. 12. The high resolution scans depict that the majority of the bonds within the materials are conjugated carbon-carbon bonds, which is expected from carbon materials. A single peak at 531.4 eV within the *O1s* resolution scan indicates that oxygen in the material is only covalently bound to carbon, with no interaction with sulfur heteroatoms. Additionally, Raman spectra of the carbonized products are found in Supplementary Fig. 13. Generally, comparing the *D/G* ratio of the areas under the peaks at 1327 cm⁻¹ (*D*: disordered) and 1579 cm⁻¹ (*G*: graphitic) provides qualitative information about the graphitic nature

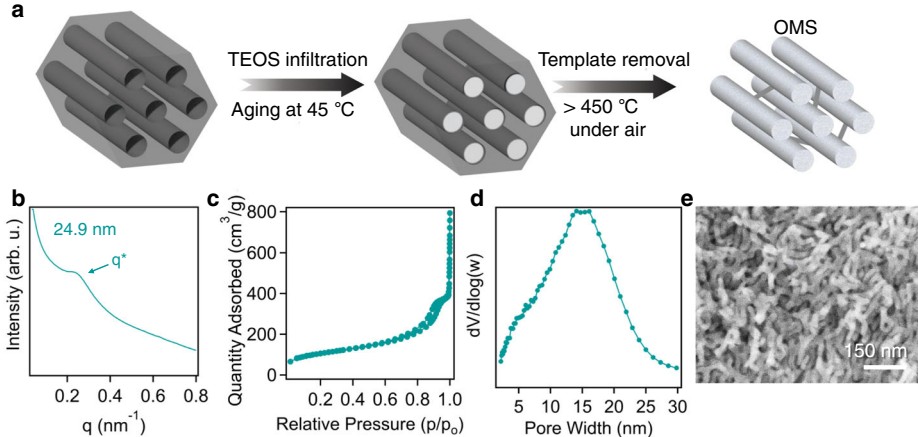

**Fig. 5 | Synthesis of OMS from SEBS-derived OMC templates. a** Schematic illustration of OMS synthesis where OMCs are infiltrated with a silica precursor which is then aged. After aging, the template is removed by exposure to high temperatures in air, resulting in mesoporous silica. (**b**) SAXS pattern, (**c**) nitrogen adsorption/ desorption isotherm, (**d**) pore size distribution, and (**e**) SEM image of OMS derived from using SEBS118–800 OMC as hard templates. Source data are provided as a Source Data file.

of the carbon. After carbonization at 800 °C, the material exhibits a *D/G* ratio of 1.73, indicating that the carbon framework is largely amorphous in nature. Previous reports incidated that polyolefin-derived carbons, crosslinked by sulfuric acid treatment, are less graphitic than their counterparts from PAN[63,64].

**Versatility of SEBS-OMC fabrication to different precursors**

The production of OMCs through the sulfonation induced crosslinking is simple and scalable, which can be extended to a broad selection of SEBS-based precursors, enabling the production of OMCs with multitudes of different pore characteristics. For instance, altering molecular weight of the constituents of the TPEs can produce OMCs with a broad range of pore sizes using the same processing methods. Three different precursors were used to demonstrate the ability of the sulfonation-induced crosslinking to convert TPEs to OMCs with varying pore textures, including SEBS89, SEBS100, and SEBS130. Information regarding their molecular weight, chemical composition, and polydispersity is included in the Fig. 4 and Methods Section. The polymers were crosslinked under identical conditions as previously demonstrated, and their representative FTIR spectra before and after 4 h of the sulfonation reaction are found in Supplementary Fig. 14. The characteristic bands corresponding to the addition of sulfonic acids to the polymer backbone, as well as the formation of double bonds are present, indicating the success of the crosslinking reaction. Additionally, TGA thermograms for each material are presented in Supplementary Fig. 15, which demonstrate residual masses in the range between 40% and 54%.

Controlling the molecular weight of the precursor provides the ability to manipulate the pore texture of the final porous product. For instance, the domain spacing of the OMC prepared using SEBS89 is reduced compared to the SEBS118-derived counterpart (33.9 nm – 24.2 nm) (Fig. 4a). This is also confirmed through the nitrogen adsorption/desorption isotherm in Fig. 4b, c, which indicates an average pore size of 10.4 nm. The surface area of this material is 216 m²/g after carbonized at 800 °C. Additionally, the SEM image in Fig. 4d confirms presence of uniform pore sizes in the carbon matrix. Figure 4e–h further demonstrates the versatility of this process through successful extension to a SEBS precursor (SEBS100) which is grafted with maleic anhydride (2 wt%). Using the same reaction conditions, SEBS100 is successfully crosslinked, as determined through FTIR spectroscopy (Supplementary Fig. 14b) and yields roughly 44% residual mass after exposure to 800 °C in a N₂ atmosphere (Supplementary Fig. 15) the carbonized products demonstrate well-ordered

pores as demonstrated through SAXS (Fig. 4e), nitrogen physisorption measurements (Fig. 4f, g), and SEM micrographs (Fig. 4h). The SEBS100-derived carbon contains pores with an average pore size of 11.3 nm and a domain spacing of 21.8 nm. This further demonstrates the tunability of the structures which can be achieved through leveraging the sulfonation induced crosslinking reaction. Additionally, it is worth noting that the crosslinking process is insensitive to the presence of the maleic anhydride functionalities grafted to the polymer backbone. Interestingly, while SEBS130 has a higher molecular weight than SEBS89, their derived OMCs still exhibit similar domain spacings (Fig. 4i), which may be attributed to (1) a lower PS volume fraction, and thus reduced degree of domain swelling upon sulfonation, and (2) a higher volume fraction of polyolefin matrix in SEBS precursors potentially leading to a larger degree of shrinkage upon conversion to OMCs. The limited swelling of PS domains in SEBS130 upon crosslinking is further evidenced by the majority pore size ranging from 4 to 5 nm (Fig. 4j–l). This result suggests that the pore wall is much thicker in the SEBS130-derived carbon, also confirmed through the SEM image in Fig. 4f. Additionally, the presence of secondary ordering peak in Fig. 4i confirms their ordered cylindrical morphology, while exhibiting an enhanced surface area of 501 m²/g. It is found that the pore size distribution of SEBS130-derived OMCs is broader than their counterpart from the other SEBS precursors, which may be attributed to the slower ordering dynamics during sulfonation-induced structural rearrangement associated with its higher molecular weight nature, and a higher degree of pore size shrinkage disrupting the morphology of resulting OMCs during carbonization process. It is also anticipated that optimizing processing conditions for each SEBS system may result in OMCs with improved degree of ordering. Nevertheless, the precursors used in this work are capable of producing OMCs with distinct pore sizes ranging from 4.7 to 16.1 nm through identical processing methods, while varying precursor identity. These results confirm the broad applicability of our process, which can be further extended to prepare a wide variety of OMCs with different pore textures by simply controlling the precursor identity, such as chemical composition and molecular weight.

**The use of SEBS-derived OMC for preparing mesoporous silica**

To further demonstrate the versatile use of large-pore OMCs from SEBS precursors, we employed SEBS118-derived OMCs carbonized at 800 °C as a hard template for ordered mesoporous silica (OMS) synthesis. The schematic illustration of this process is included in Fig. 5a, which is accomplished by backfilling the pores of SEBS-OMC

with tetraethylorthosilicate (TEOS) acting as a silica precursor. After aging in the presence of 1.6 M HCl at 45 °C for 48 h, TEOS can be crosslinked within the pores. The carbon-silica composite is dried and subsequently heated to 550 °C under air to complete remove the mesoporous carbon hard template, resulting in a relatively ordered mesoporous silica product. Figure 5b depicts the SAXS pattern of the mesoporous silica product. Compared to the OMC template, the domain spacing of OMS shrinks from 33.9 to 24.9 nm upon template removal. Additionally, the physisorption isotherm (Fig. 5c) and the associated pore size distribution (Fig. 5d) depict the presence of mesopores in the silica product with an average pore diameter of 14.7 nm. The pore size distribution is broadened in comparison to the original OMC template, suggesting the high temperature calcination step may cause nanostructure disruption. The formation of mesopores within the silica framework, through the hard-templating method, is confirmed by the SEM image in Fig. 5e. In general, commercial OMS have very limited pore sizes (less than 10 nm) due to the use of low molecular weight surfactant as the template[65,66]. Here, we have demonstrated that our fabrication method can address this challenge, leveraging low-cost SEBS derived carbons to produce large-pore OMS. Notably, XPS results indicate that the mesoporous silica is completely pure, exhibiting no trace of the carbon template after removal (Supplementary Fig. 16). Additionally, Si and O elements are present at a 1:2 atomic ratio, confirming derived products are $SiO_2$.

## Discussion

Since their inception, the development of OMCs has been heavily reliant on templating-based approaches, which require the use of templates to direct the nanostructures of precursors. While they are the most successful strategy for OMC fabrication, there are several outstanding challenges including relatively high material cost, large amounts of volatile solvent consumption, and the lack of the ability to produce large pores (e.g. >10 nm) from commercial surfactant templates. Additionally, systems which use PAN-based BCPs as precursors can exhibit limits in scalability and the ability to establish long range order in the final porous product. In this work, sulfonation-induced crosslinking of thermoplastic elastomers is demonstrated, providing a versatile platform to fabricate ordered mesoporous materials from widely available and low-cost polymer precursors, through a simple and scalable method. Specifically, the OMC production can be accomplished in two steps, eliminating the requirement of using volatile solvent for mixing in soft templating-based approaches. We would like to note that pyrolysis of crosslinked SEBS might result in sulfur and carbon emission, necessitating measures for addressing their environmental impacts. Specifically, wet flue gas desulfurization using porous alkaline-based sorbents is an established method in many industry sectors for remediating $SO_x$ gases[67].

Using the reported process, large-pore mesoporous materials can be fabricated containing a range of pore characteristics, while sulfur heteroatoms can be incorporated in the carbon framework. The pore textures and doping content can be altered by varying the processing conditions and precursor identity. Specifically, we demonstrate the capability of SEBS-derived OMCs to exhibit average pore sizes ranging from 4.7 to 16.1 nm, while the surface areas and degree of ordering of the SEBS-OMCs are reduced in comparison to other materials templated by surfactant-based molecules. Specifically, the Pluronic templates used in soft templating systems have significantly lower molecular weight than SEBS used in this work, which provides greatly enhanced mobility during the evaporation induced self-assembly process to establish well-ordered nanostructures. In this work, SEBS is reacted in the bulk and the thermodynamic drive to establish an ordered nanostructure is influenced by the increased miscibility between the blocks as they become sulfonated, and therefore their OMC structures are not as ordered as Pluronic-templated counterparts. However, the increased pore size from SEBS-derived OMC systems

could enable their use in many practical applications, which can be complementary with existing templated OMC systems. Additionally, the inherent sulfur doping can potentially enhance the functionality of the materials. Furthermore, it is important to note the utility of these materials and processing methods can also be extended to the production of large-pore ordered mesoporous silica with tailorable pore sizes which can be useful for many applications. Overall, this work provides a new and industrially feasible platform for the synthesis of mesoporous materials, which has potential toward scaled production due to its advantages of simple process and using low-cost chemicals.

## Methods

### Synthesis of OMM through sulfonation induced crosslinking of SEBS

Typically, 0.300 g of annealed SEBS118 (Sigma-Aldrich, (Molecular weight: 118,000 g/mol, Đ: 1.59, $\phi_{PS} \approx 0.20$)) and 3 g of sulfuric acid (Sigma-Aldrich, 98%) were added to a reaction vessel, along with a stir bar. Prior to this step, SEBS118 was annealed under a $N_2$ atmosphere at 160 °C for 12 h to establish ordered structures. The vessel was heated at 150 °C for varying amounts of time. After the sulfonation was complete, the contents of the reaction vessel were passed through a glass fritted funnel, and the polymer was collected. The polymer was washed with 200 mL of deionized water (Milli-Q IQ 7003 purification system, Millipore Sigma.) at least three times to completely remove the residual acid and other reaction by-products. The washed material was then dried under vacuum for their further use to produce mesoporous materials. This process was reproduced using multiple other precursors mentioned throughout this work which include: SEBS89 (Sigma-Aldrich, (Molecular weight: 89,000 g/mol, Đ: 1.56, $\phi_{PS} \approx 0.20$)), SEBS130 (Kraton), (Molecular weight: 130,000 g/mol, Đ: 1.59, $\phi_{PS} \approx 0.15$), SEBS100 (Sigma-Aldrich), (Molecular weight: 100,000 g/mol, Đ: 1.67, $\phi_{PS} \approx 0.18$) For calcination, the polymer was heated in a tube furnace (MTI Corporation OTF-1200x) under a $N_2$ atmosphere at 400 °C for 3 h with a ramp rate of 10 °C/min. The carbonized materials were also pyrolyzed in a tube furnace by first heating to 600 °C with a ramp rate of 1 °C/min followed by increasing the temperature to 800 °C, 1000 °C, or 1200 °C at a ramp rate of 5 °C/min. In the production of OMS, 0.400 g of SEBS118–800 and 0.850 g of tetraethyl orthosilicate (Sigma-Aldrich, 98%) were added to 15 g of 1.6 M HCl. The mixture was stirred at 45 °C for 48 h, then removed, dried, and heated to 550 °C for 3 h in air to remove the carbon hard template and produce the mesoporous silica product. As a note regarding safety hazards associated with this work, concentrated sulfuric acid is a strong acid and can cause harm to users upon exposure. Proper personal protective equipment must be worn to mitigate any risks while handling. Similarly, the calcination/pyrolysis process can release harmful fumes and must be carried out using a tube furnace that is properly ventilated.

### General characterization

Fourier transform infrared (FTIR) spectroscopy was performed using an attenuated total reflection FTIR spectrometer from Perkin Elmer. Spectra were recorded over a range from 4000 cm⁻¹–600 cm⁻¹ with 32 scans at a resolution of 4 cm⁻¹. Degree of sulfonation was determined by soaking ~200 mg of crosslinked polymer in 0.1 M sodium chloride (NaCl) solutions for 48 h. In doing this, ion exchange occurs between the positively charged $Na^+$ and $H^+$ ions. This solution was then titrated with 0.026 M sodium hydroxide (NaOH) solution until a pH of 7 to determine the concentration of acid within the solution and thus the amount of sulfonic acid that was present in the polymer after reaction. The degree of sulfonation was calculated using the following equation:

$$\text{Degree of sulfonation} = \frac{V_{NaOH}*M_{NaOH}}{\frac{m_{SEBS}}{M_{w,SEBS}}*N} \tag{1}$$

where $V_{NaOH}$ is the volume of NaOH required to neutralize the solution, $M_{NaOH}$ is the molarity of the NaOH solution, $m_{SEBS}$ is the mass of sulfonated polymer that was added to the NaCl solution, $M_{w,SEBS}$ is the molecular weight of the polymer (118,000 g/mol), and $N$ is the number of repeat units (~2640). Ultimately, this calculation provides the percentage of repeat units that contain sulfonic acid groups after the sulfonation reaction. Small angle x-ray scattering data was recorded at the 12-ID-B beamline at the Advanced Photon Source within Argonne National Laboratory. A q-range of approximately 0.034 nm$^{-1}$–8.8 nm$^{-1}$ was investigated using an x-ray energy of 13.3 keV. The 2D images were azimuthally averaged to provide 1-dimensional data using MATLAB programs developed in-house at 12-ID-B. All domain spacings were calculated using the relationship: $d = 2\pi/q$. 1-dimensional data were fit to models using SASview software for further data analysis. Specifically, the scattering patterns were fit to generalized scattering functions represented by the equation below:

$$I(q) = F(q)*S(q) + Bq^{-\beta} + \text{Background} \qquad (2)$$

where $I(q)$ is the intensity as a function of q-vector, $F(q)$ is the form factor, $S(q)$ is the structure factor, and $Bq^{-\beta}$ is a power law decay that accounts for contributions from interparticle scattering. A flexible cylinder form factor was used to model the shape and size of the minority phase throughout the reaction process, which has been described previously in the literature[68]. The structure factor accounts for the correlation between the minority domains[69].

Thermogravimetric analysis (TGA) experiments were conducted using a Discovery Series TGA 550 from TA instruments. Samples were heated in $N_2$ atmosphere at ramp rates of 10 °C/min. Differential scanning calorimetry (DSC) experiments were performed using a Discovery Series DSC 250 from TA instruments using a heat-cool-heat cycle to erase thermal history. The heating cycles were ramped at 10 °C/min while the cooling portions were performed with a cooling rate of 5 °C/min. Scanning electron microscopy images (SEM) were recorded on a Zeiss Ultra 60 field-emission SEM with an accelerating voltage of 17 kV and samples were carbon sputtered coated prior to imaging. Pore size analysis of SEM images was conducted using ImageJ software. X-ray photoelectron spectroscopy experiments were carried out using an ESCALAB Xi+ spectrometer (Thermo Fisher) equipped with a monochromatic Al X-ray source (1486.6 eV) and a MAGCIS Ar+/Arn+ gas cluster ion sputter gun. All spectra were recorded with a takeoff angle of 90° with respect to the surface, and the base pressure during spectral acquisition was $3 \times 10^{-7}$ mbar. High resolution scans were fit using Avantage software from Thermo Fisher. Nitrogen adsorption and desorption isotherms were recorded at 77 K through the use of a Tristar II 3020 (Micromeritics). Pore size distributions and pore volumes were calculated using non-local density functional theory (NLDFT) models for carbon slit pores at 77 K and surface areas were determined through Brunauer-Emmett-Teller (BET) analysis. Raman spectroscopy experiments were conducted using a 328i spectrometer (Andor Kymera) with 600 l/mm gratings centered around 532 nm. The system was equipped with an Andor Newton camera, and the laser was operated at 532 nm with a power of ~20 mW. Mass gain throughout sulfonation was monitored by massing the starting material prior to sulfonation and comparing to the final mass after washing and drying. Gel fraction was determined by stirring sulfonated material in toluene at 85 °C for 12 h and comparing the mass before and after.

## Data availability

The authors declare that data supporting the findings of this study are available within the paper and its Supplementary Information Files. Data generated in this study are provided in the Source Data file with this paper. Data are also available from the corresponding author upon request. Source data are provided with this paper.

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

## Acknowledgements

This project was financially supported by the start-up funding from the University of Southern Mississippi. This research used resources of the Advanced Photon Source, a U.S. Department of Energy (DOE) Office of Science user facility operated for the DOE Office of Science by Argonne National Laboratory under Contract No. DE-AC02-06CH11357. Additionally, the authors would like to thank Ivan Kuzmenko, Alexis Quental, Byeongdu Lee, Xiaobing Zuo, and Soenke Seifert for assistance during multiple X-ray scattering experiments at 12-ID-B. The authors would also like to acknowledge Dr. Derek Patton and Surabhi Jha for assisting with many XPS experiments, as well as the NSF (DMR-1726901) for providing funding for the XPS instrument. The authors thank Dr. Shan Yang for assisting Raman measurements. Finally, the authors would also like to thank Kraton for donation of polymers used in this work.

## Author contributions

M.R. and Z.Q. designed experiments for this work, where M.R. carried out the majority of them. A.G.-O., P.S., A.G., and A.B. assisted in sample preparation and data collection throughout the work. M.R. and Z.Q wrote the manuscript, which other authors also contributed to. Z.Q. supervised this project. All authors provided approval to the final version of the manuscript.

## Competing interests

Z.Q., M.R. and P.S. submitted a U.S. provisional patent for relevant technology of OMC synthesis using TPE (Serial number: 63/311,804). The remaining authors declare no competing interests.
