## [Peer Review File · Nature Communications]

Direct synthesis of ordered mesoporous materials from thermoplastic elastomersREVIEWER COMMENTS

Reviewer #1 (Remarks to the Author):

The manuscript reports a simple method that converts low-cost thermoplastic elastomers-based block copolymers in ordered mesoporous materials with tailored pore diameters. By crosslinking through sulfonation reactions of thermoplastic elastomers, sulfur can be incorporated in the carbon framework. The reported results are useful for OMC obtaining with tailored characteristics. However, the manuscript must be revised addressing the following items:

For FTIR spectra, the title of Y axis must be Absorbance (a.u.). The authors should avoid using the term of peaks in the case of FTIR spectra. Instead of peaks, use vibrations or bands. Instead of "alkyl stretches" use alkyl stretching vibrations.

It is recommended to use NLDFT model for the average pore size determination from isotherms because is more appropriate, especially for OMC.

The specific surface area must be provided as integer values because of the error of determination.

Figure 1 could be deleted because it is not very suggestive or at most integrated in figure 2

Reviewer #2 (Remarks to the Author):

In this work, the authors developed an innovative and practicable method to synthesize ordered mesoporous materials using sulfonated SEBS as model precursors via a set of crosslinking, calcination and carbonization procedures. The versatility of the materials in pore textures and doping content were demonstrated by controlling the processing conditions and precursor identity. Besides producing large pore ordered mesoporous carbons (OMS), the capability of the approach in preparing ordered mesoporous silica (OMS) with high purity is very interesting though with relative wide pore size distribution. However, the following issues need to be addressed before considering for publication.

(1) Please supplement the data of sulfonation degree of all the sulfonated SEBS used in the study.

(2) It should be noted that the content of sulfonic acid groups introduced to the polymer backbone and PS would play important roles in crosslinking degree and domain spacing of nanostructured SEBS precursor and thus is expected to be key factors in influencing the pore structure of the final mesoporous materials. However, the study on the correlation is omitted.

(3) It should be noted that the sulfonic acid groups on PS are prone to be thermally crosslinked and would participate in the formation of pore texture during the process. How about the role of this part in constructing the mesoporous materials? The relative characterization and clarification are needed.

(4) Can the data of crosslinking degree of PEB and PS blocks under different conditions be presented?

(5) The declared advantage of being low-cost and simple approach for scalable production of OMCs and OMS needs further consideration and more strict assessment considering the probable cost on controlling large amount of sulfur and carbon emission in avoiding environment impact.

Reviewer #3 (Remarks to the Author):

Using inexpensive and commercially available block copolymers and simple processes to prepare ordered mesoporous carbons is promising. The manuscript of Qiang reported a method involving sulfonation and high temperature pyrolysis to synthesis mesoporous carbons materials using commodity thermoplastic elastomers as precursors. Porous carbons with large pore size can be manufactured by employing different polymer precursors. The process can also be extended to prepare mesoporous polymer and silica. Unfortunately, the obtained materials appeared to be largely disordered, especially for these using polymers with large molecular weights (for example SEBS89, Kraton G1642). The detailed mechanism for the fabrication of ordered mesoporous structures were not discussed. This significantly reduces the novelty of the work. This work is interesting, but is not suitable for publication in Nature Communications.

1. Would the authors provide TEM images from the [110] and [001] directions to prove the ordered mesoporous materials?
2. Materials obtained with large polymers seems disordered than that of the small polymers. A detailed explanation should be provided.
3. It's confusing that the sulfur content of calcination temperate at 1200 oC (0.7 at%) was higher than that of 1000 oC (0.5 at%).
4. BJH method is not accurate and suitable to calculate the distribution of pore size larger than 10 nm.
5. Captions for Figure 4e-f were missing.

Reviewer 1

The manuscript reports a simple method that converts low-cost thermoplastic elastomers-based block copolymers in ordered mesoporous materials with tailored pore diameters. By crosslinking through sulfonation reactions of thermoplastic elastomers, sulfur can be incorporated in the carbon framework. The reported results are useful for OMC obtaining with tailored characteristics. However, the manuscript must be revised addressing the following items:

1. For FTIR spectra, the title of Y axis must be Absorbance (a.u.). The authors should avoid using the term of peaks in the case of FTIR spectra. Instead of peaks, use vibrations or bands. Instead of "alkyl stretches" use alkyl stretching vibrations.

Response: All wording in the discussion of the FTIR spectra have been edited according to the reviewer comment. We appreciate the opportunity to describe our findings more appropriately. Examples of altered text in the revised manuscript are provided below:

"At short reaction times (0.25 h, 0.5 h, and 1 h) this vibration is dominating, while the alkyl stretching vibrations associated with the PEB block at 2851 cm^{-1} and 2920 cm^{-1} only diminish slightly."

"However, the relative intensity corresponding to the alkyl stretching vibrations begins to decrease more significantly after 1 h of sulfonation, and bands associated with the addition of sulfonic acid groups (1033 cm^{-1}) and alkenes (1615 cm^{-1}) within the backbone become more present."

"The characteristic bands corresponding to the addition of sulfonic acids to the polymer backbone, as well as the formation of double bonds are present, indicating the success of the crosslinking reaction."

2. It is recommended to use NLDFT model for the average pore size determination from isotherms because is more appropriate, especially for OMC.

Response: We appreciate the opportunity to more accurately analyze the pore size distribution of mesoporous materials produced using SEBS as the precursor. We agreed that BJH method can be challenging for determining the pore size distribution (PSD) of mesoporous carbons/silica/polymers in this work. Specifically, it has been previously demonstrated that PSDs calculated for materials with large mesopores can result in additional inaccuracies due to dependencies inherent to the basis of the BJH model and the Kelvin equation. (*Appl. Catal. A Gen.*, **2006**, 313, 167-176). The first examples of using density functional theory (DFT) for pore size characterization from Seaton et al. demonstrated a more versatile approach to account for these fluctuations in pore size calculation. (*Carbon* **1989**, 27, 853-861). This work was followed by the development of non-local density functional theory (NLDFT) models which are now widely used in the characterization of mesoporous materials and is recommended as the standard method for pore size characterization by the International Standard Organization. (*Colloids Surf. A Physicochem. Eng. Asp.* **2013**, 437, 3-32. *ISO-15901-3, Pore size distribution and porosity*) All pore size distributions have now been calculated using NLDFT methods, which is demonstrated in the figure below. (a-d) are the NLDFT-derived PSDs for the calcinated sample, and the samples

carbonized at 800 °C, 1000 °C, and 1200 °C, respectively. (e-h) are the PSDs calculated from the BJH model for the corresponding samples for comparison. The averaged pore size derived using NLDFT method is closer to the pore size observations from SEM measurements, compared to using BJH models. We appreciate the reviewer for this important suggestion for improving the quality of our work.

3. The specific surface area must be provided as integer values because of the error of determination.

Response: This suggestion is now reflected in both the manuscript discussion as well as the table in supplementary information. Examples of the edited text and the revised Table S1 are provided below.

“Furthermore, BET surface areas of these OMC materials are 357 m²/g (800 °C carbonization), 404 m²/g (1000 °C carbonization), and 212 m²/g (1200 °C carbonization) (Figure 3c).”

“The surface area of this material is 216 m²/g after carbonized at 800 °C.”

“Additionally, the presence of secondary ordering peak in Figure 4e confirms their ordered cylindrical morphology, while exhibiting an enhanced surface area of 501 m²/g.”

Material	Domain Spacing (nm)	Surface Area (m ² /g)	Average Pore Width (nm)	Pore Volume (cm ³ /g)	Sulfur Content (at%)
----------	---------------------	----------------------------------	-------------------------	----------------------------------	----------------------

SEBS118-400	32.7	133	16.1	0.20	1.8
SEBS118-800	33.9	357	16.1	0.41	1.5
SEBS118-1000	29.4	404	14.7	0.38	0.9
SEBS118-1200	27.9	212	14.1	0.42	0.7
SEBS89-800	24.2	216	10.4	0.36	0.3
SEBS130-800	24.9	501	4.7	1.33	1.6
SEBS100-800	21.8	475	11.3	0.33	-
SEBS118-OMS	24.9	343	14.7	0.61	0

4. Figure 1 could be deleted because it is not very suggestive or at most integrated in figure 2

Response: We appreciate this suggestion which allows us to make our manuscript more concise. Figure 1 and Figure 2 have been combined.

Reviewer 2

In this work, the authors developed an innovative and practicable method to synthesize ordered mesoporous materials using sulfonated SEBS as model precursors via a set of crosslinking, calcination and carbonization procedures. The versatility of the materials in pore textures and

doping content were demonstrated by controlling the processing conditions and precursor identity. Besides producing large pore ordered mesoporous carbons (OMS), the capability of the approach in preparing ordered mesoporous silica (OMS) with high purity is very interesting though with relative wide pore size distribution. However, the following issues need to be addressed before considering for publication.

1. Please supplement the data of sulfonation degree of all the sulfonated SEBS used in the study.

Response: The authors agree that additional data to support the sulfonation degree will supplement the evidence of the sulfonation reaction, as well as understanding their reaction kinetics. To this end, titrations have been performed for each sample following established procedures in the literature for similar systems. (*Desalination* **2016**, 390, 33–46) To determine the sulfonation degree of the crosslinked SEBS products, the solid samples can be soaked in sodium chloride (NaCl) solutions to exchange H⁺ ions from the sulfonic acid groups with Na⁺ ions from the solution. The solution can then be titrated to determine the concentration of H⁺ which was used to calculate the sulfonation degree of SEBS samples. In our supplementary experiments, each material (SEBS118 crosslinked for different time, completely dried) was soaked in 0.1 M NaCl solutions for 48 h, and the solutions were subsequently titrated with 0.026 M sodium hydroxide (NaOH) to determine the concentration of H⁺, and thus the amount of sulfonic acid that was present in the polymer after reaction. The degree of sulfonation was calculated using the following equation:

$$\text{Degree of sulfonation} = \frac{V_{NaOH} * M_{NaOH}}{n_{sEgs} M_{wsEgs} * N}$$

Where V_{NaOH} is the volume of NaOH required to neutralize the solution, M_{NaOH} is the molarity of the NaOH solution, M_{sEgs} is the mass of sulfonated polymer that was added to the NaCl solution, M_{wsEgs} is the molecular weight of the polymer (118,000 g/mol), and N is the number of repeat units (~2640). Ultimately, this calculation provides the percentage of repeat units that contain sulfonic acid groups after the sulfonation reaction.

The degree of sulfonation for SEBS118 as a function of crosslinked time is shown below (included as Figure S4), which exhibits a similar trend to observations from FTIR, mass gain, and gel fraction experiments. The addition of the sulfonic acid to the backbone occurs steadily until roughly 3 h of reaction time. At this time, the number of sulfonic acid groups within the polymer decreases as the subsequent mechanisms in the crosslinking reaction become dominant. We have now included following descriptions and discussions in the experimental and results sections.

Figure S4. Degree of sulfonation determined through titrations of the crosslinked material as a function of reaction time.

“Additionally, titration experiments were performed on sulfonated SEBS118 samples to determine the amount of sulfonic acid groups on the polymer backbone as a function of crosslinking time. This was accomplished following an established method,³⁸ which introduced the crosslinked SEBS118 in 0.1 M sodium chloride (NaCl) solutions for 48 h to exchange the protons of the sulfonic acid with sodium ions. The solution in which the material was soaked could then be titrated using a 0.026 M NaOH solution to determine the concentration of acid present. Figure S4 depicts the sulfonation degree of SEBS118 samples as a function of crosslinking time, corresponding to the percentage of repeat units in the polymer that contain a sulfonic acid group. This result agrees well with the general trends observed in the mass gain and gel fraction studies. The sulfonation degree increases to 14% after 3 h of reaction time and then decreases slightly to 12%; further olefination and crosslinking of PEB can lead to reduced amount of the sulfonic acid groups on the polymer backbones as shown in Figure 1b. For comparison, other works which sulfonated SEBS for water desalination membranes demonstrated roughly 4% of the repeat units (including both PEB and PS segments) after 5 h of sulfonation reaction at 50 °C.³⁸ The significant increase in the sulfonation degree of SEBS resulting from our process is due to the increased reaction temperature which enables the sulfonation of both PEB and PS blocks, whereas the PEB block cannot be sulfonated at the low temperatures (e.g. ~ 50 °C).”

2. It should be noted that the content of sulfonic acid groups introduced to the polymer backbone and PS would play important roles in crosslinking degree and domain spacing of nanostructured SEBS precursor and thus is expected to be key factors in influencing the pore structure of the final mesoporous materials. However, the study on the correlation is omitted.

Response: We agree that the addition of sulfonic acid groups to the polymer backbone and the minority phase could alter morphologies and domain spacings of the samples as they undergo sulfonation. To further explore the development of the nanostructure throughout the sulfonation process, we have performed additional SAXS experiments and analyses, as shown in Figure S7a,

which is also provided below, including for polymers sulfonated at short time periods (1, 3, 5, and 10 min). By extracting the domain spacing from the peak position (Figure S7b), it is observed that the domain spacing swells from 26 nm in the neat polymer to 38 nm after 3 min of reaction. Subsequently, the domain spacing remains at relatively constant value of 38 nm throughout the reaction. This suggests that the altered nanostructure is established and kinetically trapped very early in the sulfonation process.

Figure S7. (a) SAXS patterns of sulfonated SEBS118 at short time scales at 150 °C. (b) Domain spacing determined from the primary peak in SAXS patterns as a function of reaction time. (c) Cylinder diameter determined from fitting SAXS patterns with scattering functions that include a flexible cylinder form factor which represents the size and shape of the cylindrical minority PS domains within the polymer. (d) FWHM determined from the primary peaks within the SAXS patterns to demonstrate the evolution of degree of ordering throughout the reaction process.

To further understand how nanostructures of SEBS samples evolve upon sulfonation reactions, their SAXS patterns were fit to a scattering function provided below:

$$I(q) = F(q) * S(q) + Bq^{-4} + Background$$

Where $I(q)$ is the intensity as a function of q -vector, $F(q)$ is the form factor, $S(q)$ is the structure factor, and Bq^{-4} is a power law decay that accounts for contributions from interparticle scattering. A flexible cylinder form factor was used to model the shape and size of the minority phase throughout the reaction process, which has been described previously in the literature. (*Macromolecules* **1996**, 29, 7602–7612) The structure factor accounts for the correlation between the minority domains. (*J. Chem. Phys.* **1998**, 79, 2461) All model fitting was performed using SASview data analysis software. The results of the model fitting are provided in Figure S7c. Specifically, the diameter of the cylinder which represents the PS minority domains was extracted from the flexible cylinder form factor. The cylinder diameter increases rapidly from 16.6 nm to 22.0 nm within 3 min of reaction. This value increases slightly through the reaction process to 24.0 nm. These results further confirm that the nanostructure is established very quickly during the crosslinking reaction at 150 °C. Notably, the increase in cylinder diameter (5.4 nm) suggests the domain spacing increase of SEBS (12 nm) was primarily contributed from PS domain expansion. Additionally, the limited structure rearrangement of SEBS samples after crosslinking for 3 min is

associated with reduced polymer mobility upon sulfonation. Particularly, the presence of charged groups (i.e. sulfonic acids) on PS and PEB can significantly hinder chain dynamics due to ionic associations serving as physical crosslinkers, which has been reported in many literatures (*Macromolecules* **1991**, 24, 1071-1076; *Polymer*, **2004**, 45, 3001-3016; *Polymer*, **2001**, 42, 2321-2330; *Rheologica Acta*, **2021**, 60, 241-249; *Phys. Rev. Lett.*, **2016**, 116, 158001). Therefore, even though the chemical reactions of PS and PEB progress, the structure is kinetically trapped after a short period of crosslinking time.

Furthermore, the degree of ordering of the system was also evaluated by calculating the full width at half maximum (FWHM) of the primary scattering peaks for each sample (Figure S7d). After 1 min of reaction, the FWHM increased significantly in comparison to the neat polymer, which can be attributed to the diffusion-enabled, heterogeneous SEBS crosslinking reactions. After 3 min, the FWHM decreases to a value similar to that of the neat polymer. In contrast to the domain spacing and cylinder radius, the FWHM gradually increases throughout the reaction process, indicating a slightly reduced degree of ordering. This could be a result of increasing degrees of sulfonation and the diffusion of sulfuric acid into the polymer improving the miscibility between PEB and PS blocks of the polymer.

Moreover, the implications of the nanostructure throughout the process on the pore texture of the final product were investigated by carbonizing the samples which were sulfonated for shorter periods of time and performing SAXS and nitrogen physisorption experiments. Additionally the adsorption isotherms and corresponding pore size distributions (Figure S9a,b) demonstrate similar results, but an increase in the averaged pore size from 14.1 nm to 15.4 and 15.6 nm was observed for samples crosslinked for 1 h, 2 h and 3 h, respectively. The sample crosslinked for 4 h has an averaged pore size of 16.1 nm. While the morphology of the sulfonated precursor is very similar, the alteration in pore structure can be explained through the TGA thermogram in Figure S9c. The samples sulfonated for shorter periods of time undergo much more significant degradation after exposure to elevated temperatures, which could result in further reduced pore sizes (due to higher degree of shrinkage from lower crosslinking degree) and lower degrees of carbon yield. Importantly, these values were maximized for after 4 h of sulfonation, which was the reaction condition used throughout the rest of the work.

Figure S9. (a) Nitrogen physisorption isotherms and (b) pore size distributions of SEBS118-derived OMC which were sulfonated for 1 h, 2 h, and 3 h, respectively. (c) TGA thermograms of sulfonated SEBS118 illustrating the dependence of carbon yield on reaction time.

The following discussions have now been included in the revised manuscript:

“Interestingly, we found that the nanostructure of SEBS is altered almost immediately upon exposure to the sulfonating agent at 150 °C, evidenced by a rapid increase in their domain spacing. Specifically, Figure S7a depicts SAXS patterns of samples at early times in the reaction progression, from 1 min to 10 min of reaction. The domain spacing increases rapidly after 3 min of reaction to 38 nm and remains virtually constant throughout 4 h of reaction (Figure S7b). Additionally, the scattering patterns were fit to model scattering functions which included a flexible cylinder form factor to account for scattering contributions from the size and shape of the minority cylindrical PS domains. As shown in Figure S7c, a similar trend to the domain spacing evolution was observed where the cylinder diameter increased rapidly at short time scales from 16.6 nm to 22.0 nm within 3 min of reaction, and then gradually increased throughout the reaction to 24.0 nm. These results further confirm that the nanostructure is established at very short reaction times and is only altered slightly at extended reaction times. Notably, comparing the increase in cylinder diameter throughout the reaction (7.4 nm) to the increase in domain spacing (12 nm) suggests that PS domain expansion as a result of the sulfonation reaction is the primary contributor to the altered nanostructure. As the sulfonation reaction progresses, PEB crosslinking and the presence of ionic groups on polymer backbones can significantly hinder the polymer chain mobility for structural rearrangement, kinetically trapping the morphology of SEBS after relatively short sulfonation. It is noted that the structures tend towards a slightly reduced degree of ordering with extended reaction times. The full width at half maximum (FWHM) of the primary peak within the scattering patterns was analyzed and provided in Figure S7d. Within the first minute of reaction, the peak broadens significantly as rearrangement of the nanostructure occurs. Subsequently, the FWHM decreases sharply after 3 min of reaction as long-range ordering is reestablished. In contrast to the domain spacing and cylinder diameter which remained nearly constant as the reaction progressed, a slight increase is observed in the FWHM with SEBS sulfonation progressing.”

“It is worth noting that the reaction time directly impacts the nanostructure of the porous carbon product. Shorter sulfonation times are still sufficient for producing relatively well-ordered porous carbon materials as illustrated in Figure S9. Specifically, nitrogen physisorption isotherms (Figure S9a) and pore size distributions further confirm the presence of ordered pore structures (Figure S9b). Samples which were crosslinked for 1 h, 2 h, and 3 h demonstrated a gradual increase in the averaged pore size from 14.1 nm (sulfonating for 1 h) to 15.6 nm (sulfonating for 3 h). These results suggest that SEBS-derived OMC can have process-tunable pore textures, enabling controlled resulting pore sizes by varying crosslinking conditions. The TGA thermogram in Figure S9c reveals that increased reaction times are required for maximizing the carbon yield of the

material after carbonization. Samples sulfonated for 1 h exhibited only 12% yield (from sulfonated samples) after exposure to 800 °C under N₂, which increased to 26% and 34% after reaction for 2 h and 3 h, respectively. After 4 h, the yield is maximized at 42%, and this condition was considered optimal for further studies.”

3. It should be noted that the sulfonic acid groups on PS are prone to be thermally crosslinked and would participate in the formation of pore textile during the process. How about the role of this part in constructing the mesoporous materials? The relative characterization and clarification are needed.

Response: We thank the reviewer for raising this point. While it is true that the sulfonic acid groups were chemically attached to the PS segments upon sulfonating SEBS samples, the resulting bond is still thermally labile and prone to degradation. The authors have provided a TGA thermogram of homopolymer PS sulfonated under the same conditions, which demonstrates no carbon yield after exposure to 800 °C in N₂ atmosphere, in comparison to neat SEBS and the crosslinked SEBS material. This suggests that the PS minority phase in the SEBS block copolymer will not contribute to the carbon yield of the final product. Previous work associated with sulfonated PS also suggests their very minimal carbon yield up to 800 °C (*J. Mater. Chem. A*, 2013, 1, 13989). Additionally, several works in the literature focus on implementing the sulfonic acid crosslinking method for functionalizing aromatic polymers such as poly(ether ether ketone), towards various applications. Among these studies, (*Renewable Sustainable Energy Rev.*, 2021, 137, 110471) the complete thermal degradation at 800 °C of the sulfone bridges formed during crosslinking of sulfonated aromatic polymers was demonstrated. (*J. Phys. Chem. B*, 2009, 113, 7505-7512, *Chem. Mater.*, 2006, 18, 69-75) Specifically, Di vona et al. investigated the mechanism of the thermally driven formation of sulfone crosslinks between sulfonic acids in sulfonated PEEK, and the effect of a thermal treatment on the crosslink density of the material. (*Fuel Cells*, 2013, 13 107-117, <https://doi.org/10.1002/FUCE.201200010>.) This work demonstrated that across many thermal treatments, which could be seen as analogous to the crosslinking reaction used for SEBS in this work, that no considerable residual mass was present after exposure to 800 °C in N₂.

Furthermore, an additional illustrative TGA is provided below. The SEBS118 sample was reacted at 85 °C for 12 h which is sufficient to sulfonate both the PS and PEB domains of the polymer without resulting in crosslinking, as demonstrated by FTIR. In the highlighted region of the spectrum, there is evidence of the sulfonation of both the polymer backbone (1033 cm⁻¹) and the aromatic ring of polystyrene (1006 cm⁻¹). This is further confirmed by the broad band ranging from 1293 cm⁻¹ to 1111 cm⁻¹. This band is a convolution of contributions from different chemical functionalities including sulfonic acids and sulfones. Even though the SEBS is sulfonated at low temperatures, the ability of the sulfonic acid groups to crosslink is not sufficient to produce carbon yield.

We have added following discussions in our revised manuscript as shown below:

“Notably, the sulfonation reaction occurs at the aromatic ring of the PS repeat units, and sulfonic acid groups could undergo crosslinking with other sulfonic acid moieties to produce intermolecular sulfone bridges at elevated temperatures.⁴¹ Although this effectively crosslinks the polymers, these crosslinking sites are still thermally labile and decompose at elevated temperatures rather than producing carbon. For instance, Di vona et al. have investigated the mechanism of thermally driven sulfone crosslinking between sulfonic acid moieties in sulfonated poly(ether ether ketone) (PEEK).⁴² Across many thermal treatments of the sulfonated PEEK materials, that could be seen as analogous to the crosslinking reaction used in this work, no considerable residual mass was present after exposure to 800 °C in N₂. This suggests that, while the PS domains (~20 vol% in SEBS118) may undergo intermolecular crosslinking through this mechanism, it is still insufficient to produce carbon and will not contribute to the carbon yield of the OMCs. To further confirm these results, SEBS118 was sulfonated at 85 °C for 12 h to sulfonate both the PEB and PS blocks of the polymer without inducing crosslinking. The FTIR spectrum in Figure S8a depicts bands associated with the sulfonation of both PEB and PS blocks (bands at 1033 cm⁻¹ and 1006 cm⁻¹, respectively). Additionally, a broad band ranging from 1293 cm⁻¹ to 1111 cm⁻¹ is a convolution of vibrations associated with sulfonic acids, as well as sulfones. However, despite the sulfonation of these materials, the TGA thermogram in Figure S8b indicates that no carbon is yielded after carbonization at 800 °C in a N₂ atmosphere, suggesting that any crosslinking that sulfonated PS segments did not produce carbons upon pyrolysis. ”

4. Can the data of crosslinking degree of PEB and PS blocks under different conditions be presented?

Response: The authors agree that illustrating the effects of varying sulfonation process parameters can better support the fabrication methods demonstrated in the manuscript. The same experiments were carried out at both 85 °C and 125 °C for different times to illustrate the necessity of elevated temperatures for the crosslinking reaction. Mass gains for both processing conditions are significantly lower than that of the fabrication method presented in the manuscript. Additionally, the highest gel fraction of the 85 °C and 125 °C samples was ~12% and ~60%, respectively, suggesting that the degree of crosslinking is lower compared to their counterparts at 150 °C. This is also demonstrated in the characteristic FTIR vibrations which have been illustrated in the manuscript. The following excerpt was added to the manuscript:

“To illustrate the importance of the elevated reaction conditions used in this process, the sulfonation reaction was also performed at 85 °C and 125 °C for up to 6 h. Mass gain, gel fraction, and FTIR results are presented in Figure S6. Both temperatures exhibit slower kinetics than the sulfonation at 150 °C and also demonstrate lower plateau values of mass gain and gel fraction. Sulfonation at 85 °C and 125 °C only achieve ~40% mass gain over 6 h compared to 60% for SEBS118 sulfonated at 150 °C. Similarly, the gel fractions of SEBS118 sulfonated at 85 °C and 125 °C were approximately 12% and 60%, respectively. The reduced gel fraction in comparison to the 150 °C reaction condition suggests that lower temperature sulfonation reactions result in reduced degrees of crosslinking. Furthermore, the FTIR spectra in Figure S6b and d indicate a reduced presence of the characteristic bands (sulfonic acids: 1033 cm^{-1} and 1006 cm^{-1} , alkenes: 1615 cm^{-1}) associated with the crosslinking reaction, in addition to the retention of the alkyl stretching vibrations (2851 cm^{-1} and 2920 cm^{-1}) indicating a not fully completed reaction. This suggests that sulfonation temperature of SEBS is an important process parameter to control their crosslinking kinetics.”

Figure S6. (a) Mass gain and gel fraction of SEBS118 sulfonated at 85 °C as a function of sulfonation time and (b) FTIR spectrum of SEBS118 sulfonated at 85 °C for 6 h. (c) Mass gain/gel fraction of SEBS118 sulfonated at 125 °C as a function of reaction time and (d) FTIR spectrum of SEBS118 reacted at 125 °C for 6 h.

5. The declared advantage of being low-cost and simple approach for scalable production of OMCs and OMS needs further consideration and more strict assessment considering the probable cost on controlling large amount of sulfur and carbon emission in avoiding environment impact.

Response: The authors agree with this point regarding the accurate use of words and statements. For the time being, the wording has been removed to mention only the potential for having these advantages. Specifically, we now only refer "low-cost" to the commodity SEBS, instead of our manufacturing process. Additionally, we have included some additional discussions about remediating gaseous by-products in our process.

"We would like to note that, pyrolysis of crosslinked SEBS might result in sulfur and carbon emission, necessitating measures for addressing their environment impacts. Specifically, wet flue gas desulfurization using porous alkaline-based sorbents is an established method in many industry sectors for remediating SO_x gases. (*Sep. Purif. Technol.* **2022**, 281, 119849)."

Reviewer 3

Using inexpensive and commercially available block copolymers and simple processes to prepare ordered mesoporous carbons is promising. The manuscript of Qiang reported a method involving sulfonation and high temperature pyrolysis to synthesis mesoporous carbons materials using commodity thermoplastic elastomers as precursors. Porous carbons with large pore size can be manufactured by employing different polymer precursors. The process can also be extended to prepare mesoporous polymer and silica. Unfortunately, the obtained materials appeared to be largely disordered, especially for these using polymers with large molecular weights (for example SEBS89, Kraton G1642). The detailed mechanism for the fabrication of ordered mesoporous structures were not discussed. This significantly reduces the novelty of the work. This work is interesting, but is not suitable for publication in Nature Communications.

Response: We appreciate the reviewer for providing useful comments and feedback. In addition to responding to the specific comments listed below, we would like to clarify our systems and potential advantages, toward broadly benefiting scientific community by demonstrating a simple, widely applicable OMC synthesis using commodity plastic materials. We also performed significant amount of additional experiments to understand nanostructure formation mechanism during sulfonation, crosslinking/sulfonation kinetics, impact of processing conditions on OMC pore textures, as well as further extending our method to another SEBS precursor to confirm generalizability. To provide a brief summary, we found that sulfonation of SEBS yielded a rapid increase in the domain size within few minutes, and this morphology was then immediately trapped due to significantly hindered chain dynamics upon attaching sulfonic acid group, which can form ionic associations serving as physical crosslinkers. Additionally, by varying crosslinking

time, averaged pore size of resulting OMCs can be altered from 14.1 nm (1h sulfonation at 150 °C) to 16.2 nm (4h sulfonation at 150 °C), while the carbon yield was also changed. These results suggest the opportunity to control pore size by processing conditions in our SEBS-derived OMC systems. Additionally, we found decreasing reaction temperature will yield a reduced gel content in crosslinked SEBS. Through these experiments, we were able to understand the nanostructures and chemistry of SEBS and their derived porous materials at each processing stage. By including a new SEBS precursor, we also found that the wide pore size distribution of SEBS130 may be attributed to the combined effects of slower ordering dynamic during sulfonation-induced structural rearrangement associated with higher molecular weight nature, and a higher degree of pore size shrinkage disrupting the morphology of resulting OMCs during carbonization process. We believe these additional results and discussions further strengthened the quality of our work.

Moreover, it is worth discussing that the materials produced in this work are inherently less ordered than mesoporous carbons produced through soft-templating based techniques, such as the Pluronic-templated mesoporous carbons. The Pluronic templates used in soft templating have significantly lower molecular weight than SEBS samples used in this work. This provides greatly enhanced system mobility during the evaporation induced self-assembly process to establish well-ordered nanostructures. However, the low molecular weight leads to their limited attainable pore size window, which is typically up to 10 nm. The need of preparing large-pore mesoporous carbons using a simple method is apparent (as suggested by several great review and perspective articles, *Chem. Soc. Rev.* **2013**, 42, 4054, *J. Am. Chem. Soc.*, 2017, 139, 5, 1706-1713, *Angew Chem Int Ed*, **2008**, 47, 3696-3717; *Nat. Comm*, **2020**, 11, 4984), which is difficult to access using conventional Pluronic templating systems. In this work, SEBS is reacted in the bulk and the thermodynamic drive to establish an ordered nanostructure is influenced by the competition between increasing compatibility between the blocks as they become sulfonated, and therefore their OMC structures are not as ordered as Pluronic-templated counterparts. However, in comparison to mesoporous carbons produced from the direct pyrolysis of PAN-containing precursors (the most common, widely used system to date), the materials produced through our method are much more ordered (*J. Am. Chem. Soc.*, **2012**, 134, 14846-14857, *Nature Comm.*, **2019**, 10, 675), which is evidenced by much narrower primary ordering peak in SAXS patterns as well as narrower pore size distribution. A direct comparison of the order in carbon produced from the SEBS precursors to the PAN-derived OMCs in the previously mentioned papers is provided below:

Editorial Note: Panel below reprinted with permission from Zhong et al., Electrochemically Active Nitrogen-Enriched Nanocarbons with Well-Defined Morphology Synthesized by Pyrolysis of Self-Assembled Block Copolymer, *Journal of the American Chemical Society* **2012** 134 (36), 14846-14857. Copyright 2012 American Chemical Society.

Nature Comm., 2019, 10, 675

J. Am. Chem. Soc., 2012, 134, 14846-14857

This work

Specifically, the primary scattering peaks are much sharper and more distinct, indicating a more highly ordered structure than the provided examples. This is a result of the narrow processing window of PAN limited by a high melting point, high T_g , and a crosslinking temperature that occur within a narrow temperature range. Herein, we addressed this challenge by using SEBS, which has an amorphous and low T_g polyolefin matrix.

We would like to clarify that using PAN-based BCPs for synthesizing OMCs has several important advantages, including resulting in nitrogen-doped carbon framework, self-crosslinking without the need of additional crosslinking agents, which are different from our system. However, it is important to highlight that SEBS is a commodity thermoplastic elastomer. For example, all SEBS samples employed in our work are industrial grade SEBS (with relatively high polydispersity) which are produced at a very large scale, providing another important advantage for enabling the transformative impact of our reported system. Collectively, we believe this work is important to provide a new synthetic route for preparing large-pore, sulfur-doped OMCs using low-cost chemical agents. We have provided relevant discussions in our revised manuscript.

“The surface areas and degree of ordering of the SEBS-OMCs are reduced in comparison to other materials templated by surfactant-based molecules. Specifically, the Pluronic templates used in soft templating systems have significantly lower molecular weight than SEBS used in this work, which provides greatly enhanced mobility during the evaporation induced self-assembly process to establish well-ordered nanostructures. In this work, SEBS is reacted in the bulk and the thermodynamic drive to establish an ordered nanostructure is influenced by the competition between increasing compatibility between the blocks as they become sulfonated, and therefore their OMC structures are not as ordered as Pluronic-templated counterparts. However, the

increased pore size from SEBS-derived OMC systems could enable their use in many practical applications, which can be complementary with existing templated OMC systems.”

1. Would the authors provide TEM images from the [110] and [001] directions to prove the ordered mesoporous

Response: We agreed that additional TEM images would further confirm the ordered mesoporous structures of OMCs, supplementing existing SAXS, pore size distribution, and SEM data. We have now included TEM micrographs of SEBS118-derived OMC carbonized at 800 °C (corresponding to Figure 3f).

Figure S11. TEM images of SEBS118-800 along the (a) [110] and (b) [001] directions.

2. Materials obtained with large polymers seems disordered than that of the small polymers. A detailed explanation should be provided

Response: We thank reviewer for this comment and appreciate this opportunity for further understanding and clarifying our systems.

First, for both materials (SEBS89 and SEBS130, which were named as SEBS89 and Kraton G1642 in the first version of our manuscript), their SAXS patterns and nitrogen physisorption isotherms (and derived pore size distribution) indicate the presence of ordered nanostructures in the carbon product. However, their original SEM images appear more disordered than OMC from the other polymer precursor. This is likely a result of issues that we had with initially resolving the structures with smaller pore sizes. To remediate this, we carbon coated the materials prior to imaging and improved the ability to image at higher resolutions due to improved imaging contrast. New images using this technique are now provided in the revised manuscript.

Second, we agree that OMC derived from SEBS130 (molecular weight of 130,000 g/mol) exhibits a wider pore size distribution than from SEBS89 (molecular weight of 89,000 g/mol), suggesting that the precursor composition has an impact on the degree of ordering of their resulting OMCs. We attribute the lower degree of ordering in SEBS130-derived OMCs to 1) their polymer chain dynamics for structural re-arrangement, 2) high degree of pore size shrinkage. SEBS with higher molecular weight polymers have a lower mobility for developing their ordering during sulfonation

reactions, and they might be kinetically trapped very quickly due to synergistic effects of crosslinking, high molecular weight, and presence of charged groups. Additionally, we note that pore shrinkage is significant in resulting OMC (from SEBS130) due to their relatively low PS volume fraction in the precursor, which may also result in a more disrupted nanostructure after carbonization, associated with wider pore size distribution. To further study how polymer molecular weight impacts resulting OMC structures, another SEBS precursor (SEBS100, with a molecular weight of 100,000 g/mol) was employed and studied. We found that this precursor can result in OMC with a domain spacing of 21.8 nm (Figure 4i) and an averaged pore size of 11.3 nm (Figure 4j and k). SEM image also further confirm the presence of relatively well-ordered cylindrical pores (Figure 4l). The degree of ordering is similar to the OMC from SEBS89 and SEBS118. These results further confirm the precursor molecular weight and composition could have a direct impact on the OMC structures, which can be explained by associated polymer chain mobility for ordering and structural developing during crosslinking reaction.

The following responses has been included into the discussion within the manuscript:

“Figure 4e-h further demonstrates the versatility of this process through successful extension to a SEBS precursor (SEBS100) which is grafted with maleic anhydride (2 wt%). Using the same reaction conditions, the carbonized products demonstrate well-ordered pores as demonstrated through SAXS (Figure 4e), nitrogen physisorption measurements (Figure 4f,g), and SEM micrographs (Figure 4h). The SEBS100-derived carbon contains pores with an average pore size of 11.3 nm and

a domain spacing of 21.8 nm. This further demonstrates the tunability of the structures which can be achieved through leveraging the sulfonation induced crosslinking reaction. Additionally, it is worth noting that the crosslinking process is insensitive to the presence of the maleic anhydride functionalities grafted to the polymer backbone.”

“Similarly, SEBS130-derived carbon exhibits smaller pore sizes than their counterparts from SEBS118 and SEBS89. Interestingly, while SEBS130 has a higher molecular weight than SEBS89, their derived OMCs still exhibit similar domain spacings (Figure 4i), which may be attributed to 1) a lower PS volume fraction, and thus reduced degree of domain swelling upon sulfonation, and 2) a higher volume fraction of polyolefin matrix in SEBS precursors may lead to a larger degree of pore shrinkage upon conversion to OMCs. The limited swelling of PS domains in SEBS130 upon crosslinking is further evidenced by the majority pore size ranging from 4 - 5 nm (Figure 4j-l). This result suggests that the pore wall is much thicker in the SEBS130 derived carbon, which is also confirmed through the SEM image in Figure 4f. Additionally, the presence of secondary ordering peak in Figure 4i confirms their ordered cylindrical morphology, while exhibiting an enhanced surface area of 501 m²/g. It is found that the pore size distribution of SEBS130-derived OMCs is broader than their counterpart from the other SEBS precursors, which may be attributed to the slower ordering dynamic during sulfonation-induced structural rearrangement associated with high molecular weight nature, and a higher degree of pore size shrinkage disrupting the morphology of resulting OMCs during carbonization process.”

3. It’s confusing that the sulfur content of calcination temperate at 1200 °C (0.7 at%) was higher than that of 1000 °C (0.5 at%).

Response: The authors appreciate the reviewer for questioning this point. We believe that these two measurements (with slight difference) were slightly skewed by localized information during XPS survey scans. To ensure the validity of the measurements, the same samples have been scanned in triplicate and averaged. The results are provided below. According to updated characterization experiments, we have now revised our manuscript.

“Increasing carbonization temperature to 1000 °C and 1200 °C decreases the sulfur content to 0.9 at% and 0.7 at% as the heteroatoms are eliminated from the framework at elevated temperatures.”

Trial	1000 °C			1200 °C		
	C (%)	O (%)	S (%)	C (%)	O (%)	S (%)
1	93.6	5.1	0.9	96.0	3.3	0.6
2	92.5	6.6	0.9	95.8	3.6	0.6
3	91.4	7.6	1.0	96.7	2.5	0.7
AVG	92.5	6.4	0.9	96.2	3.1	0.7

4. BJH method is not accurate and suitable to calculate the distribution of pore size larger than 10 nm.

Response: We are thankful for the opportunity to characterize our materials more appropriately. It has been demonstrated throughout the literature that the BJH method can misrepresent pore sizes within mesoporous materials, especially those with larger pores. This is a result of inherent dependencies within the Kelvin equation which acts as the basis for the BJH model. Since the initial development of density functional theory, great strides have been made towards the adaptation of these models for the characterization of pore size distributions within porous materials. Non-local density functional theory (NLDFT) models are now widely used for the characterization of mesoporous materials and are much more versatile than some classical methods. NLDFT models now widely used in the characterization of mesoporous materials and is recommended as the standard method for pore size characterization by the International Standard Organization. (*Colloids Surf. A Physicochem. Eng. Asp.* **2013**, 437, 3–32. *ISO-15901-3, Pore size distribution and porosity*) The pore size distributions have now been calculated using NLDFT models which are provided below and have been added into the supporting information. Specifically, the models are for slit pores with N₂ adsorbate at 77 K. Additionally, this has been added to the experimental section of the manuscript. The averaged pore size derived using NLDFT method is closer to the pore size observations from SEM measurements, compared to using BJH models.

5. Captions for Figure 4e-f were missing.

Response: The authors appreciate the reviewer for catching this oversight, and the mistake has been fixed in the text of the manuscript.

"SEM micrographs of (e) mesoporous polymer and mesoporous carbon carbonized at (f) 800 °C, (g) 1000 °C, (h) 1200 °C."

REVIEWERS' COMMENTS

Reviewer #1 (Remarks to the Author):

The authors improved their manuscript addressing all issues pointed out. The manuscript can be published in this form.

Reviewer #2 (Remarks to the Author):

I have carefully read the response and revised manuscript. I am pleased with the well addressed answer and clear explanation to each question based on implementing sufficiently supplemental experiment and investigation. The quality of the present work has been significantly improved. I recommend the publication of the work in Nature Communications.

Reviewer #3 (Remarks to the Author):

The authors have provided some new results to prove the ordered nanostructures of the carbon materials. Additional experiments have been also carried out to understand nanostructure formation mechanism. The quality of this work have been improved. I have no more questions.

Responses to reviewers' comments

Reviewer #1 (Remarks to the Author):

The authors improved their manuscript addressing all issues pointed out. The manuscript can be published in this form.

Reviewer #2 (Remarks to the Author):

I have carefully read the response and revised manuscript. I am pleased with the well addressed answer and clear explanation to each question based on implementing sufficiently supplemental experiment and investigation. The quality of the present work has been significantly improved. I recommend the publication of the work in Nature Communications.

Reviewer #3 (Remarks to the Author):

The authors have provided some new results to prove the ordered nanostructures of the carbon materials. Additional experiments have been also carried out to understand nanostructure formation mechanism. The quality of this work have been improved. I have no more questions.

Response: We thank for the favorable considerations from all reviewers. We appreciate their time and efforts for providing comments, which are very valuable for improving the quality of our work.